

# *De novo* assembly and characterization of the transcriptome of *Morchella esculenta* growth with selenium supplementation

Mengxiang Du[1,*], Shengwei Huang[2,*], Zihan Huang[2], Lijuan Qian[1], Yang Gui[2], Jing Hu[2] and Yujun Sun[2]

[1] College of Agriculture, Anhui Science and Technology University, Fengyang, Anhui, China

[2] School of Life and Health Science, Anhui Science Technology University, Fengyang, Chuzhou, Anhui, China

[*] These authors contributed equally to this work.

## ABSTRACT

Although *Morchella esculenta* (L.) Pers. is an edible and nutritious mushroom with significant selenium (Se)-enriched potential, its biological response to selenium stimuli remains unclear. This study explored the effect of selenium on mushroom growth and the global gene expression profiles of *M. esculenta*. While 5 μg mL$^{-1}$ selenite treatment slightly promoted mycelia growth and mushroom yield, 10 μg mL$^{-1}$ significantly inhibited growth. Based on comparative transcriptome analysis, samples treated with 5 μg mL$^{-1}$ and 10 μg mL$^{-1}$ of Se contained 16,061 (452 upregulated and 15,609 downregulated) and 14,155 differentially expressed genes (DEGs; 800 upregulated and 13,355 downregulated), respectively. Moreover, DEGs were mainly enriched in the cell cycle, meiosis, aminoacyl-tRNA biosynthesis, spliceosome, protein processing in endoplasmic reticulum pathway, and mRNA surveillance pathway in both selenium-treated groups. Among these, MFS substrate transporter and aspartate aminotransferase genes potentially involved in Se metabolism and those linked to redox homeostasis were significantly upregulated, while genes involved in isoflavone biosynthesis and flavonoid metabolism were significantly downregulated. Gene expression levels increased alongside selenite treatment concentration, suggesting that high Se concentrations promoted *M. esculenta* detoxification. These results can be used to thoroughly explain the potential detoxification and Se enrichment processes in *M. esculenta* and edible fungi.

# INTRODUCTION

As an essential micronutrient for humans and animals, selenium (Se) is crucial for cellular physiological processes, including cellular antioxidant defenses, regulating thyroid hormone metabolism, and cell growth. Daily selenium supplementation is therefore important for human health (*Fan et al., 2024*; *Kieliszek, 2019*; *Razaghi et al., 2021*). However, selenium has harmful and beneficial attributes. Selenium intake exceeding the 400 μg daily limit may cause vomiting, diarrhea, abdominal pain, fatigue, hair loss, joint pain, or increase incidence of prostate cancer or risk of type 2 diabetes (*Hadrup & Ravn-Haren, 2020*; *MacFarquhar et al., 2010*; *Spiller & Pfiefer, 2007*; *Thompson et al., 2016*).

Corresponding author
Yujun Sun, sunyujun208@163.com

Therefore, people must pay more attention to dietary selenium consumption to avoid selenium-deficient defects and potential overdose risks (*Rayman et al., 2018*; *Thompson et al., 2016*).

Selenium's chemical structure and dosage predicate its biological activity, metabolism, toxicity, and potential applications. Inorganic selenium forms (primarily selenite and selenate) can be biotransformed into organic forms (selenomethionine, selenocysteine, and other seleniocompounds) through incorporation into amino acids or polysaccharides (*Qian et al., 2023*). These organic selenium compounds present higher bioaccessibility, reduced toxicity, and more beneficial effects than inorganic forms (*Ragini & Arumugam, 2023*; *Shao et al., 2021*). Given selenium and selenoproteins' high nutritional value, selenium biofortification using agricultural products grown in bioaccumulative organisms is a promising alternative (*Ávila et al., 2013*; *Businelli et al., 2015*). Particularly, Se-enriched wheat, onions, cabbage, broccoli, pumpkin, chives and garlic have been recommended as Se-fortified food crops or sources of selenium-enriched food supplements (*Pyrzynska, 2009*).

Compared to cultivated plants, edible mushrooms are valued for their chemical and nutritional properties, texture, and flavor (*Tsai, Tsai & Mau, 2008*). Moreover, edible mushrooms exhibit high Se bioaccumulating potential in their fruiting bodies and have been proven as promising alternatives for Se bioaccumulation and transformation (*Dong, Xiao & Wu, 2021*; *Niedzielski et al., 2015*; *Rathore et al., 2018*). For example, *Flammulina velutipes* can accumulate nearly 108 $\mu g\,g^{-1}$ of organic Se under treatment with 20 $\mu g\,g^{-1}$ selenite, accounting for over a 97% intake of the total selenium (*Dong, Xiao & Wu, 2021*). Moreover, other mushrooms, including *Hypsizygus marmoreus*, *Hericium erinaceus*, *Ganoderma lucidum*, and *Auricularia cornea* are known to be effective selenium accumulators and good supplements (*Hu et al., 2021*; *Li et al., 2019*; *Xu et al., 2021a*; *Xu et al., 2021b*). Moreover, the synergistic effect of organic selenium compounds and other active ingredients induces potent chemopreventive activities (*Sun et al., 2017*).

Despite their use as a dietary supplement in China and other countries, the metabolism and responses of selenium-fortified mushrooms are poorly understood. Selenite's effect on mushroom fruiting and mycelial development and its adverse impact from toxic quantities have been documented in various mushrooms (*Song et al., 2022*; *Wang et al., 2016*). For example, addition of 1.0 $\mu g\,g^{-1}$ selenite in the medium resulted in up to 25% increase in mushroom yield of *F. velutipes*, while 50 $\mu g\,g^{-1}$ selenite treatment significantly inhibit the growth of fruiting body (*Dong, Xiao & Wu, 2021*). Furthermore, some studies noted discrepancies in the ability of disparate mushroom species to accumulate and utilize selenite (*de Oliveira, Naozuka & Landero-Figueroa, 2022*; *Dong, Xiao & Wu, 2021*). The variations in absorption and utilization efficiency of different mushroom fruiting bodies may be attributed to the mushroom's cytoreductive system and the metabolic pathways associated with selenium metabolism (*Wang et al., 2016*; *Zhang et al., 2022b*; *Zhao et al., 2020*). For example, selenium uptake by *F. velutipes* decreased by 25 or 26% upon phosphate or sulfite starvation, respectively (*Wang et al., 2016*), while Se-enriched *Pleurotus citrinopileatus* displayed significantly higher sulfur metabolism and ABS transporter activity (*Zhao et al., 2020*). At the same time, there have been a number of studies on the biological

processes and reduction mechanisms about Se transformation in edible mushrooms. Transcriptional profiling of *Auricularia cornea* under selenium treatment showed that selenium accumulation increased genes linked to glutathione metabolisms, translation, and other metabolic processes (*Li et al., 2019*). Furthermore, tandem mass tag-based quantitative proteomic analysis revealed that thioredoxin 1, thioredoxin reductase (NADPH), glutathione reductase, and cystathionine gamma-lyase were key proteins involved in selenite reduction and methylation in *G. lucidum* (*Xu et al., 2023*). Thus, distinct transporters and metabolic pathways may mediate and produce different forms of selenium compounds in mushrooms. However, research on these specific mechanisms requires further study.

*Morchella esculenta* (commonly known as yellow morels) is one of the most consumed mushrooms in the world, especially in Europe, the USA, India, and China (*Heleno et al., 2013*; *Maryam Ajmal et al., 2015*), and is appreciated for its delicacy, specific texture, and aroma (*Dospatliev et al., 2019*; *Qian et al., 2023*). *M. esculenta* is rich in many health-promoting nutritional components (*Kewlani et al., 2023*; *Vassilev, Denev & Papazov, 2020*). Current research has highlighted that *M. esculenta*-derived compounds possess excellent biological activities, including antibacterial (*Haq et al., 2022*), antioxidant (*Akyuz et al., 2019*; *Bedlovičová et al., 2024*), and anti-inflammatory (*Zhang et al., 2022a*; *Zhang et al., 2022b*). For instance, polysaccharide (Se-MPS) prepared from selenium-enriched *M. esculenta* could significantly enhance the activity and phagocytosis of macrophages and secrete pro-inflammatory cytokines to exert immunomodulatory functions compared to ordinary morel polysaccharides (*Qian et al., 2023*). Therefore, Se-enriched *M. esculenta* represents an alternative source for nutritional Se enrichment and a commercially viable or marketable product.

Selenium uptake, tolerance, and metabolism pathways in *M. esculenta* were investigated to better understand the effect of toxic and low Se concentrations on *M. esculenta* and the mechanisms underlying Se accumulation. This study investigated selenium's impact on mushroom growth in *M. esculenta* through treatments with different amounts of selenium. Moreover, the transcriptome of *M. esculenta* at two specific selenite concentrations (low and high) was explored to uncover the dose-specific mechanisms of selenium enrichment and to provide a theoretical framework for future research on the metabolism and selenium enrichment of mushrooms.

## MATERIALS & METHODS

### Strain and culture conditions

The *Morchella esculenta* (L.) Pers. strain (ACCC50764) was provided by the Agricultural Culture Collection of China (Beijing, China) and was cultured on potato dextrose agar (PDA) plates (Qingdao Hopebio Technology, Qingdao, China). Solid and liquid cultures were performed in an ordinary PDA liquid medium (without sodium selenite) or Se-PDA liquid medium (supplementation with sodium selenite) at $26 \pm 0.5$ °C based upon experimental need.

## Chemicals and reagents

Sodium selenite ($Na_2SeO_3$, $\geq$ 99%) was provided by Sigma-Aldrich (St Louis, MO, USA). The other chemicals used (such as nitric acid, hydrochloric acid, potassium borohydride, and absolute ethanol) were of analytical grade purity and purchased from Sangon Biotech Co., Ltd. (Shanghai, China) or Aladdin Chemistry Co., Ltd. (Shanghai, China).

## Se tolerance and growth rate/biomass measurement

Solid-state fermentation using Se-PDA or ordinary PDA plates was employed to determine growth rate and biomass production. $Na_2SeO_3$ was diluted with distilled water and added to modified PDA to establish the final selenium concentrations ranging from 1 to 30 $\mu g\,mL^{-1}$. The colony diameter was measured twice, each perpendicular to the other, on the third and seventh days to evaluate the growth rate (mm $d^{-1}$). To determine the effect of selenium treatment on the colony color, the colony color was observed against a white background. Liquid cultures were initiated by transferring three 6-mm-diam agar plugs into 15 mL of PDA liquid medium in 125 mL Erlenmeyer flasks at 26 $\pm$ 0.5 °C for two days. Next, biomass production was assessed by injecting 15 mL of the *Morchella* liquid strand to infect 150 mL of PDA liquid medium containing different $SeO_3^{2-}$ concentrations (0.05 to 30 $\mu g\,mL^{-1}$). The liquid culture was performed according to our previously published work (*Qian et al., 2023*). After 10 min of centrifugation (10,000 rpm), the mycelium was harvested and placed in an oven at 60 °C until its weight remained constant. The total Se levels within the mycelium for each treatment sample were ascertained through hydride generation-atomic fluorescence spectrometry (HG-AFS), as described by *Wu et al. (2020)*.

## Preparation of mycelial samples for transcriptome sequencing

First, 5 $\mu g\,mL^{-1}$ and 10 $\mu g\,mL^{-1}$ were selected as the promoting and inhibiting Se concentrations based on the selenite growth response to investigate the Se-enrichment process of *M. esculenta*. Following cultivation on PDA plates at 26 $\pm$ 0.5 °C for three days, the mycelial of the *M. esculenta* strain and the PDA media were cut into 1 cm $\times$ 1 cm pieces and placed into 250 mL flasks filled with PDA liquid medium containing 0, 5, and 10 $\mu g\,mL^{-1}$ sodium selenite (labeled as CK, MSe5, and MSe10, respectively). Each sample had three replicates. After incubation at 26 $\pm$ 0.5 °C for seven days, the mycelial sample (1 g wet weight) for each treatment was harvested through filtration and stored at −80 °C until RNA extraction.

## Library construction and RNA-Seq

RNA was extracted using a Total RNA Extraction Kit per the manufacturer's instructions (Solarbio Science and Technology Co., Ltd., Beijing, China). The quantify the total RNA was measured by 2100 Bioanalyzer (Agilent Technologies, Santa Clara, CA, USA). Nine cDNA libraries were constructed using the TruSeq RNA Sample Preparation Kit (Illumina, San Diego, CA, U.S.A.) after RNA quality assessment and contaminated RNA removal. The quality of the cDNA libraries was also measured using the 2100 Bioanalyzer (Agilent Technologies, Santa Clara, CA, USA). The libraries were divided into three groups (CK, MSe5, and MSe10, respectively) based on the Se concentration of the treatment groups, each containing three replicates. Each cDNA library was pair-end sequenced on an Illumina

NovaSeq 6000 sequencer at Shanghai BIOZERON Co., Ltd (Shanghai, China). Clean readings from raw data were obtained by filtering out ambiguous reads with more than 5% unknown bases (''N'') or duplicate reads from PCR after sequencing.

### *De novo* transcriptome assembly and gene functional annotation

Trinity, a short-read assembly program designed for transcriptome analysis, was used for *de novo* assembly based on clean reads (*Grabherr et al., 2011*). Low-quality (Q <20) data and adapter sequences were removed from the analysis. The statistical power of this experimental design, calculated in RNASeqPower is 0.88. Next, Bowtie2 (ver. 2.2.5) was used to extract mitochondrial and non-coding RNAs from trimmed pair-end reads (*Langmead & Salzberg, 2012*). Every cluster's longest sequence was chosen as a single unigene and sequentially annotated by comparing its transcripts with those found in Swissprot, Kyoto Encyclopedia of Genes and Genomes (KEGG) (http://www.genome.jp/kegg), Gene Ontology (GO) (http://wego.genomics.org.cn), and The National Center for Biotechnology Information's (NCBI) NR/Nt databases. Finally, GO and KEGG pathways were used to evaluate DEGs, as described by *Lu & Bau (2017)*.

### Identification of differentially expressed genes

The fragments per kilobase of exon region in a given gene per million mapped fragments approach (FRKM) was applied to calculate differentially expressed genes (DEGs) of different libraries using the DESeq R package (DESeq version: 1.30.0, R version: 4.0.4). DEGs with a significant difference were filtered using the following threshold: $|log2(FoldChange)|>1$ and $p$-value <0.05. In addition, two-way hierarchical clustering (complete linkage) was performed among the selenium-treated (MSe5 and MSe10) and control groups by adopting the R package Pheatmap to demonstrate the expression pattern of each DEG across all samples (*Metsalu & Vilo, 2015*). Moreover, DEGs were subjected to KEGG pathway enrichment analysis and GO functional enrichment using a previously published formula (*Kim et al., 2020*). A $p$-value cutoff of less than 0.05 was used to evaluate the significance of each TOG or KEGG pathway's enrichment pattern between the selenium-treated and control groups.

### Differentially expressed gene validation using real-time quantitative PCR

Real-time quantitative PCR(RT-qPCR) was used to validate the seven randomly selected DEGs and transcriptome sequencing data reliability. Primer Premier 5.0 was used to design gene-specific primers for the chosen unigene sequences; the primer sequences are listed in Table S2. The *M. esculenta* strain was cultured under conditions similar to those for RNAseq library construction, as previously described. PCR reactions were performed using the one-step RT-qPCR kit (Sangon Biotech Co., Ltd., Shanghai, China) per the manufacturer's instructions. The $2^{-\Delta\Delta Ct}$ method was utilized to determine the target gene's relative expression level. The 18S rRNA was used as housekeeping gene and the expression levels of specific DEGs were normalized through comparison with the internal reference gene.

**Table 1  The effect of sodium selenium on the growth and morphology of _M. esculenta_.**

| Selenite concertation ($\mu$g mL$^{-1}$) | Colony color | Average growth rate of mycelium (mm d$^{-1}$)[*] | Biomass (g L$^{-1}$)[*] |
|---|---|---|---|
| 0 | White | $22.33 \pm 0.635a$ | $5.27 \pm 0.203a$ |
| 1 | White | $23.51 \pm 0.208ab$ | $5.38 \pm 0.089a$ |
| 5 | White | $25.25 \pm 0.291b$ | $5.53 \pm 0.142a$ |
| 10 | White | $19.13 \pm 0.521c$ | $4.82 \pm 0.24a$ |
| 20 | Reddish | $11.72 \pm 0.186d$ | $3.75 \pm 0.181b$ |
| 30 | Reddish | $10.32 \pm 0.289d$ | $2.43 \pm 0.185c$ |

**Notes.**
    [*]Mean $\pm$ standard error.

# RESULTS AND DISCUSSION

## _M. esculenta_ growth and morphology under selenium accumulation

Mycelia morphologies in selenite-supplicated media at concentrations ranging from 1 to 5 $\mu$g mL$^{-1}$ were similar to those in the control group. However, visible indicators were noted with Na$_2$SeO$_3$ concentrations over 10 $\mu$g mL$^{-1}$. The mycelia transitioned from a white to a reddish color (Fig. 1); red indicated elemental Se (Se$^0$) formation rather than organic selenium compounds in the mycelia (_Xu et al., 2021a_). Moreover, the growth rate of mycelium and _M. esculenta_ biomass under different selenium treatments varied significantly by Se concentration (Table 1). The initially augmented mycelia growth was inhibited by an increase in Na$_2$SeO$_3$ concentration. For example, adding small amounts of selenite (5 $\mu$g mL$^{-1}$) to the medium slightly promoted mycelia growth, resulting in a 4.93% increase in mushroom yield (Table 1). However, 10 $\mu$g mL$^{-1}$ selenite treatments significantly reduced mycelial development and biomass compared with the control group. These results corroborate those reported in other studies (_Kim et al., 2014_; _Song et al., 2009_; _Xu et al., 2021a_). For example, lower Se concentrations of 1, 10 and 100 $\mu$M significantly improved the biomass levels of the fungus _Pleurotus eryngii_, while higher Se concentrations of 1,000 and 10,000 $\mu$M can significantly inhibit the mycelial growth (_Kim et al., 2014_). Furthermore, Se concentrations of 0–80 $\mu$g mL$^{-1}$ promoted mycelial growth of _Stropharia rugoso-annulata_, whereas the mycelia biomass decreased rapidly when the concentration exceeded 200 $\mu$g mL$^{-1}$ (_Song et al., 2009_). Different selenium concentrations may influence mushroom growth due to the toxic effects caused by selenium exposure. Se exposure can induce toxicity and alter the growth of organisms by causing lipid peroxidation, producing reactive oxygen species (ROS), and diminishing sulfur metabolism (_Kieliszek et al., 2020_; _Li et al., 2007_; _Staneviciene et al., 2023_). Thus, the mushroom's tolerance and capacity to metabolize selenium may be related to its species and antioxidant system. At the same time, the life cycles and substrate composition of different edible mushrooms may also account for the differences in selenite tolerance.

## _M. esculenta_ transcriptome sequencing and _de novo_ assembly

Gene expression profiles of _M. esculenta_ were compared between 5 $\mu$g mL$^{-1}$ (MSe5) and 10 $\mu$g mL$^{-1}$ (MSe10) selenium treatments and the control (CK) using high-throughput

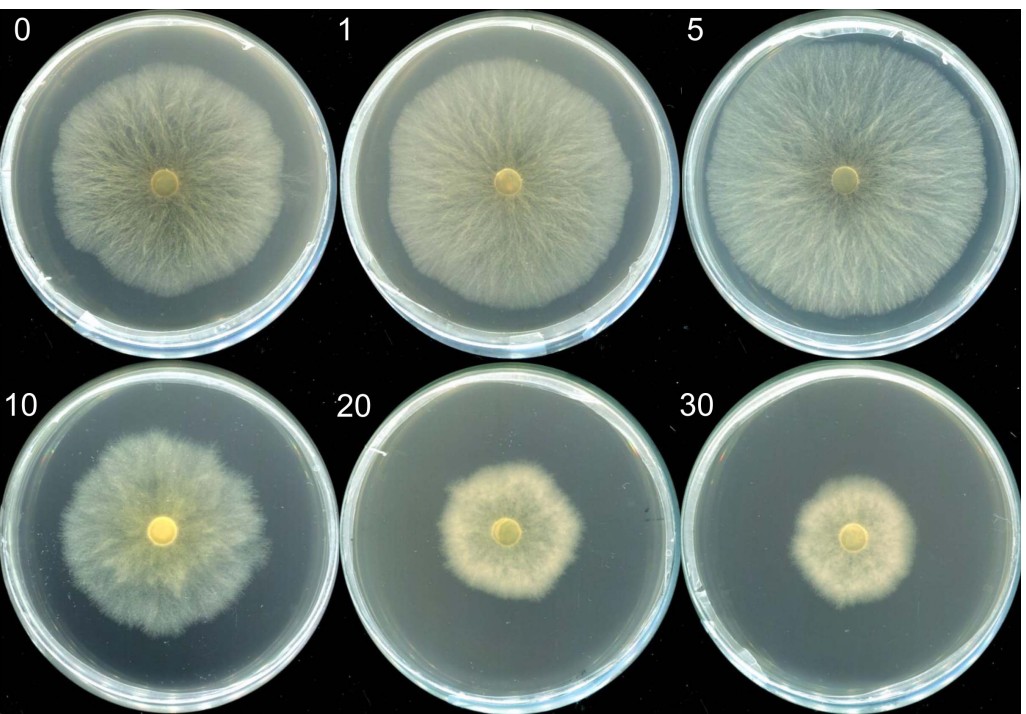

**Figure 1 The morphologies of mycelia of *M. esculenta* grown on media containing different concentrations of selenite.** The morphologies of mycelia of *M. esculenta* grown on media containing selenite ranging from 1 to 30 μg mL$^{-1}$.

RNA sequencing based on the Illumina HiSeq4000 platform. Since *M. esculenta*'s genomic data has yet to be reported, *de novo* assembly was selected for this study. An average of $2.76\times10^7$ clean readings per sample were obtained after quality examination and filtration (Table S1). Thereafter, 73,135 transcripts and 73,135 unigenes were created by assembling the clean readings. Unigene lengths ranged from 227 to over 46,000 bp, averaging 2,819 bp. The number and quality score of aligned reads were generally sufficient for subsequent transcriptome analysis.

## Transcriptional data analysis

First, 73,135 transcripts from the Trinity assembly were screened for subsequent DEG identification. For samples treated with 5 μg mL$^{-1}$ of selenium (MSe5), 452 upregulated genes and 15,609 downregulated genes were identified (Fig. 2A). Following 10 μg mL$^{-1}$ selenium treatment (MSe10), 14,155 genes were statistically upregulated (800 DEGs) or downregulated (13,355 DEGs) relative to the controls (Fig. 2B). Furthermore, co-expressed DEGs were identified using a Venn diagram and the results revealed that 475 and 2,381 genes were specifically expressed in the two comparison groups (MSe10 *vs.* CK, MSe5 *vs.* CK) and 13,680 DEGs were shared (Fig. 3). The relative expression of seven randomly selected DEGs was determined through RT-qPCR to verify the RNA-seq results (Table S2).

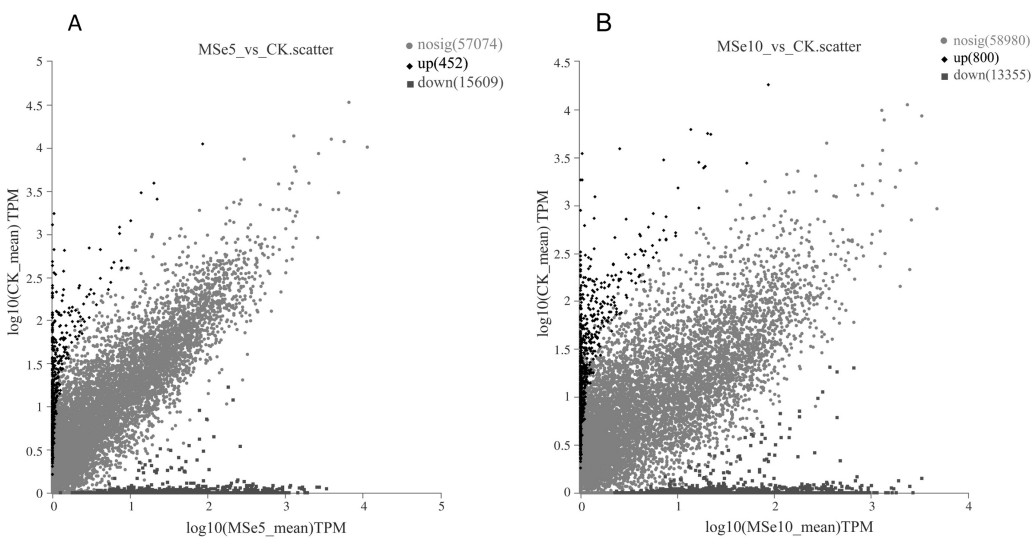

**Figure 2** **Volcano plot of DEGs between (A) MSe5 *vs.* CK and (B) MSe10 *vs.* CK groups.** The diamond, circle and square pots indicate differentially upregulated, differently downregulated, and non-differentially expressed genes, respectively. Each point in the diagram represents a gene.

Overall, regulation of the tested genes was consistent with the transcriptome analysis results, indicating that the latter was reliable.

GO enrichment analyses demonstrated that these DEGs could be classified into three categories: cellular components (CC), biological processes (BP), and molecular functions (MF). In the MSe5 *vs.* CK group, the upregulated genes of cellular components category were primarily part of the membrane's cellular components (111 DEGs) and cell parts (93 DEGs). Regarding biological processes, upregulated genes were implicated in metabolic processes (85 DEGs), cellular processes (85 DEGs), and biological regulation (25 DEGs). Furthermore, upregulated genes classified into the molecular function category were associated with binding (143 DEGs), catalytic activity (120 DEGs), and transporter activity (24 DEGs) (Fig. 4A, Table S1).

Comparatively, 15,609 downregulated genes were primarily associated with biological processes, cellular components, and molecular functions (see Table S2). The downregulated genes within biological processes primarily contributed to cellular processes (4,252 DEGs), metabolic processes (3,548 DEGs), and biological regulation (1,314 DEGs). The most apparent differences concerning the downregulation of genes within the cellular component category were those related to cell parts (4,106 DEGs), membrane parts (2,517 DEGs), and organelles (2,077 DEGs). Downregulated genes that enriched molecular functions participated in catalytic activity (5,538 DEGs), binding (5,145 DEGs), and transporter activity (575 DEGs) (Fig. 4B).

Likewise, the 800 upregulated DEGs in the MSe10 *vs.* CK group concerning biological processes predominantly contributed to cellular processes (168 DEGs), metabolic processes (163 DEGs), biological regulation (38 DEGs), and localization (33 DEGs) (Fig. 4C). DEGs that enriched cellular components were primarily associated with membrane parts (191

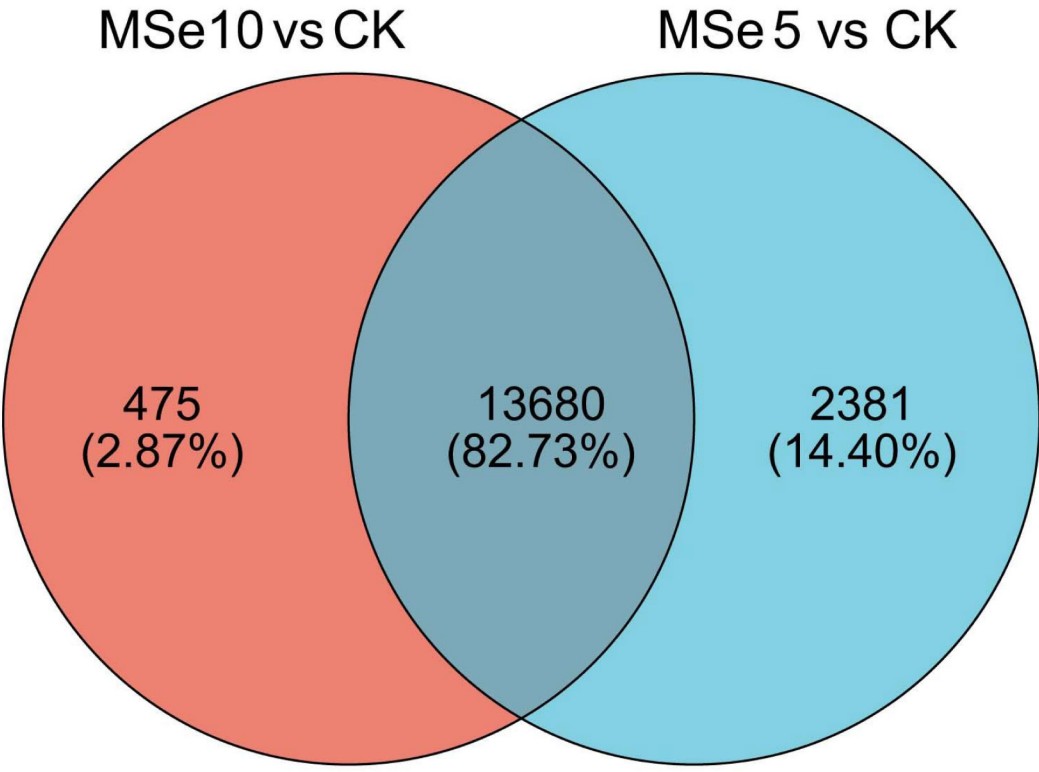

**Figure 3** Venn diagram of shared DEGs between the different selenium treatment groups MSe5 *vs.* CK and MSe10 *vs.* CK.

DEGs) and cell part (156 DEGs), while the DEGs within molecular functions were those largely linked to catalytic activity (269 DEGs) and binding (214 DEGs) (see Table S3). Furthermore, downregulated gene participation in biological processes included cellular processing (3,818 DEGs), metabolic processing (3,201 DEGs), and biological regulation (1,163 DEGs) (Fig. 4D). The most prevalent differences of MF involved catalytic activity (4,894 DEGs), binding (4,572 DEGs), and transporter activity (501 DEGs). In addition, downregulated genes concerning cellular components contributed to cell parts (3,725 DEGs), membrane parts (2,209 DEGs), organelles (1,883 DEGs), and protein-containing complexes (1,615 DEGs) (see Table S4).

The GO functional annotation results indicated that selenium enrichment or treatment can regulate numerous genes related to energy production/metabolism, protein synthesis, cell wall organization or biogenesis, and carbohydrate catabolism (*Jiao et al., 2022*; *Zhou et al., 2018*). Notable, the mostly highly enriched GO terms between the MSe5 *vs.* CK and MSe10 *vs.* CK groups were identical, suggesting that genes in these clusters were clearly Se-related. For example, DEGs participated in biological regulation or binding were all highly enriched in the MSe10 *vs* CK group and MSe5 *vs.* CK group. Similar to our results, *Li et al. (2023)* found that feeding selenium-enriched yeast-treated diets can significantly regulate the expression of genes involved in metabolism-related biological processes, such as biological regulation, molecular processes in Broiler chickens (*Gallus*

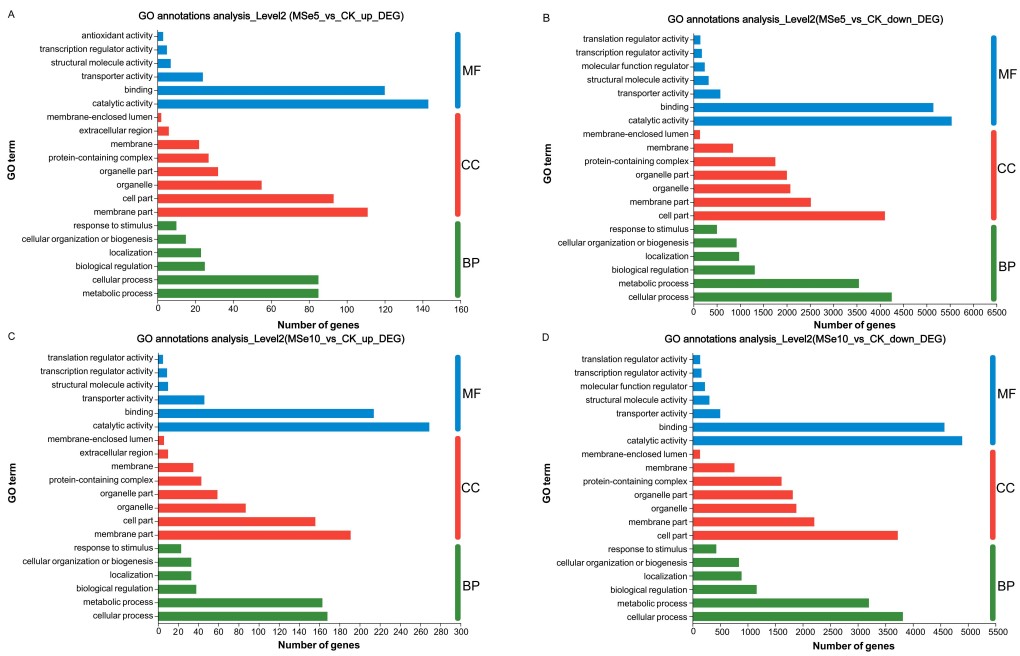

**Figure 4** **GO functional classification of *M. esculenta* unigenes different groups.** GO functional classification of *M. esculenta* unigenes in (A) Upregulated genes in MSe5 *vs.* CK, (B) Downregulated genes in MSe5 *vs.* CK, (C) Upregulated genes in MSe10 *vs.* CK, and (D) Downregulated genes in MSe10 *vs.* CK groups. The *X*-axis depicts the number of DEGs, and the *Y*-axis correlates to various gene functions. CC, cellular components; BP, biological processes; MF, molecular functions.

*gallus*). While Se treatment can modulate the global transcriptome of Yorkshire-Landrace gilts. The significantly enriched GO terms in selenite feeding group are mainly related with protein binding and translation initiation factor activity (especially translation factor activity-RNA binding) (*Dalto et al., 2018*). Meanwhile, higher selenium treatments upregulated more genes than lower concentration treatments. Other studies reported similar findings regarding selenium enrichment in *Cardamine hupingshanensis* (*Zhou et al., 2018*), and *Bombyx mori* (*Jiang et al., 2020*). For example, in the hyperaccumulator plant *C. hupingshanensis*, more DEGs (42 upregulated and 61 downregulated) with various physiological processes and cellular functions were apparent following high Se concentration treatment (80,000 μg Se L$^{-1}$). Contrastingly, only 23 upregulated and 7 downregulated genes were found in the low Se (100 μg Se L$^{-1}$) treatment group (*Zhou et al., 2018*).

## KEGG pathway enrichment analysis

KEGG was used to systematically investigate the DEGs in different Se concentration treatments to elucidate their potential functions in metabolic pathways. Notably, complex pathways were involved in the selenium stress response when treated with different selenium concentrations. In the MSe5 *vs.* CK group, upregulated and downregulated genes were enriched in 59 and 119 pathways, respectively. As shown in Fig. 5A, the enriched KEGG pathways for the upregulated DEGs in the MSe5 *vs.* CK group were primarily involved in

the cell cycles (11 DEGs), meiosis (10 DEGs), aminoacyl-tRNA biosynthesis (six DEGs), nucleocytoplasmic transport (six DEGs), glycine, serine, and threonine metabolism (four DEGs); and penicillin and cephalosporin biosynthesis (two DEGs). Meanwhile, the KEGG pathways enriched with downregulated DEGs included spliceosomes (222 DEGs), mRNA surveillance pathway (137 DEGs), protein processing in the endoplasmic reticulum (212 DEGs), glycolysis/gluconeogenesis (126 DEGs), and pyruvate metabolism (105 DEGs) (Fig. 5B). Likewise, upregulated DEGs in the MSe10 *vs.* CK group can be divided into 81 known pathways, such as meiosis (16 DEGs), the cell cycle (17 DEGs), cofactor biosynthesis (15 DEGs), pentose and glucuronate interconversions (six DEGs), the longevity regulating pathway (seven DEGs), the MAPK signaling pathway (seven DEGs), and aminoacyl-tRNA biosynthesis (seven DEGs) (Fig. 5C). Comparatively, most downregulated DEGs were associated with spliceosome (206 DEGs), protein processing in the endoplasmic reticulum (196 DEGs), the mRNA surveillance pathway (119 DEGs), glycolysis/gluconeogenesis (114 DEGs), and ribosomes (252 DEGs) (Fig. 5D). Transcriptome analysis of *Aegilops tauschii* Coss. grown under different Se treatments revealed some pivotal pathways that may participate in Se metabolism including genes involved glutathione metabolism, phenylpropanoid biosynthesis, and aminoacyl-tRNA biosynthesis (*Wu et al., 2019*). Quantitative proteomic analysis of *Ganoderma lucidum* cultured with 200 ppm selenite showed that a number of metabolic pathways were prominently enriched such as pyruvate metabolism, citrate cycle, tyrosine metabolism, and glycolysis/gluconeogenesis (*Xu et al., 2023*). Therefore, KEGG enrichment analysis indicated that these genes may play roles on Se accumulation or tolerance in *M. esculenta*. It is noteworthy that several DEG-enriched pathways in the MSe5 *vs.* CK and MSe10 *vs.* CK groups were identical, primarily encompassing cell growth and death (21 DEGs), translation (17 DEGs), aging (five DEGs), and carbohydrate metabolism (four DEGs). Similar results were also observed following selenium accumulation in the *Camellia sinensis* tea plant. Under selenite treatment, DEGs mainly enriched in ribosomes (52 unigenes) and oxidative phosphorylation (14 unigenes) were upregulated in the roots and leaves (*Cao et al., 2018*). In the fat body of silkworms after 50 µM and 200 µM selenite treatment, DEGs were predominantly associated with the KEGG pathways, including hypertrophic cardiomyopathy, dilated cardiomyopathy, and the VEGF signaling pathway, and salivary secretion (*Jiang et al., 2020*). The consistent expression of genes under different treatment conditions indicates that selenite treatment activated genes in these pathways.

## Genes involved in Se enrichment or biotransformation

Uptake is the first step for cells to assimilate and utilize selenium; thus, efficient transport systems are crucial for selenium uptake (*Wang, Rensing & Zheng, 2022a*). Several transporters involved in Se uptake and translocation have been identified (*Wang et al., 2024*). The monocarboxylate transporter Jen1p and phosphate transporters Pho87P/Pho90P/Pho91P, which facilitated selenite with high affinities, are responsible for selenite accumulation in yeast (*McDermott, Rosen & Liu, 2010*). In addition, selenite can be transported by several bacteria, including *Escherichia coli* and *Proteus* sp. YS02., *via* the sulfate ABC transporter complex (*Rosen & Liu, 2009*; *Wang et al., 2022b*). On the other

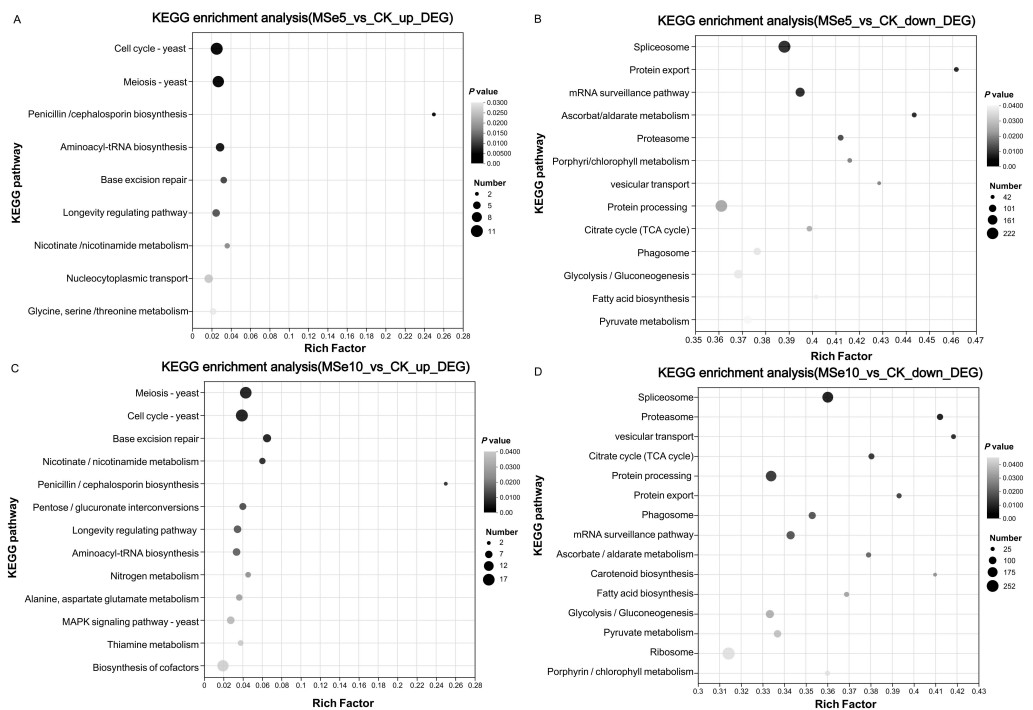

**Figure 5  The KEGG enrichment analysis of DEGs within different *M. esculenta* groups based on enrichment factors.** (A) The top nine pathways of upregulated genes in MSe5 *vs.* CK; (B) The top 13 pathways of downregulated genes in MSe5 *vs.* CK; (C) The top 13 pathways of upregulated genes in MSe10 *vs.* CK; (D) The top 15 pathways of downregulated genes in MSe10 *vs.* CK. The horizontal axis represents the name of the enriched KEGG pathway, the vertical axis represents the enrichment factor of significantly different genes in that pathway, the dot color represents the Q-value, and the dot size represents the number of genes.

hand, selenite transporters within edible fungi have not been studied at the molecular level. This study found no significant changes in gene expression, such as sulfate transporters or phosphate transporters, but revealed that the MFS substrate transporter (NODE_1033) was significantly upregulated following selenite treatment (MSe5 *vs.* CK: 5.15 $Log_2FC$, MSe10 *vs.* CK: 5.66 $Log_2FC$). The upregulation of MFS transporter gene expressions has been reported in selenium-enriched lactic acid bacteria (*Liao & Wang, 2022*). Thus, the continuous enhancement of MFS substrate transporter (NODE_1033) gene expressions in this study indicated that non-specific transporter proteins may carry Se oxyanions in the culture medium into the *M. esculenta* cells.

Moreover, gene expressions related to redox homeostasis, such as genes encoding thioredoxin and glutathione S-transferase (GstA), peroxisome, and glutathione synthetase, were significantly expressed following treatment with different selenite concentrations (*Jiang et al., 2020*; *Wang et al., 2023*). For example, the $Log_2FC$ of genes encoding thioredoxin and glutathione S-transferase were 10.2 and 9.27 in the MSe5 *vs.* CK group, respectively. In the MSe10 *vs.* CK group, the $Log_2FC$ of these gene expressions increased to 10.92 and 10.49, respectively. These findings indicate that as selenite treatment

concentrations increase, the gene expressions are also upregulated. In comparison, genes encoding glutamate-cysteine ligase (GCS) were only significantly expressed in the MSe10 *vs.* CK group (Log$_2$FC = 7.16).

Reactive oxygen species (ROS) can damage cell membranes (*Lu et al., 2022*). Thioredoxin (TRX) and glutathione are integral for ROS detoxification during stress responses and for determining the thiol/disulfide status of proteins by sensing and reducing equivalents to many target proteins, including reductases and peroxidases (*Zhou et al., 2018*). It is commonly known that oxidative stress predicates selenium cytotoxicity. If the selenium concentration in the environment is too high, selenium stress can activate the antioxidant system to scavenge ROS, enhancing the resistance of microbes or plants to oxidative stress (*Luo et al., 2021*; *Varlamova et al., 2021*). Therefore, increasing the expression of these genes demonstrated that these proteins are crucial for maintaining the redox balance in *M. esculenta* cells under selenium stress, establishing cellular resistance to selenium.

Mushrooms absorb selenium from the soil or culture medium and convert it into organic Se (primarily selenoproteins and selenopolysaccharides) (*de Oliveira, Naozuka & Landero-Figueroa, 2022*; *Qian et al., 2023*). Se methylation and selenocysteine (SeCys) generation are the two main stages of selenoprotein synthesis. Selenite treatment upregulated gene expressions involved in cysteine and methionine metabolisms, whereas those enriched in cysteine synthesis pathways were significantly downregulated. For example, the expression of genes encoding aspartate aminotransferase, which possess cysteine sulfinate and aspartate aminotransferase activities, were significantly upregulated following 10 μg mL$^{-1}$ selenite treatment with a Log$_2$FC value of 5.5. This result is consistent with previous findings concerning selenium accumulation in other eukaryotic organisms (*Cao et al., 2018*; *Zhang et al., 2019*). For instance, genes involved in the metabolism of sulfur, cysteine, and methionine were markedly upregulated in the roots of the tea plant *Camellia sinensis* after selenite treatment and are probably linked to Se accumulation (*Cao et al., 2018*). Under acid stress, genes related to ATP production and sulfur metabolism were significantly upregulated, including methionine, cysteine, and GSH production in selenium-enriched *Candida utilis* (*Zhang et al., 2017*).

Previous studies have discovered that the total flavonoid and chlorogenic acid contents of several high plants can be increased if the applied dose of selenium does not exceed a certain concentration (*Guo et al., 2020*; *Li et al., 2020*). Therefore, the expression profiles of structural genes associated with isoflavones and polysaccharides were examined to determine whether sodium selenite treatment would enhance the biosynthesis of either compound in *M. esculenta*. The results indicated that two genes related to isoflavone biosynthesis or regulation, eight related to flavonoid metabolism, and five in biosynthetic pathways were downregulated (See Table S5 and Table S6). For example, the expression of genes encoding phosphoglucose isomerase, which directs Glc-6-phosphate conversion, was significantly downregulated following 5 μg mL$^{-1}$ and 10 μg mL$^{-1}$ selenite treatment (Log$_2$FC = -13.53 for both). Furthermore, the expression of genes encoding flavanone 4-reductase/Bifunctional dihydroflavonol 4-reductase, involved in flavonoid metabolism, was significantly downregulated at 10 μg mL$^{-1}$ selenite treatment (Log$_2$FC = −9.6) but showed no effect following 5 μg mL$^{-1}$ selenite treatment. These results are consistent

with previous study indicating that an application dose exceeding 2.0 mg kg$^{-1}$ impairs total flavonoid and chlorogenic acid levels (*Wang et al., 2021*). Furthermore, an initial rise of total polyphenol contents/total flavonoid content followed by a reducing trend was detected in selenium biofortified germinated black soybeans with the increases in Se concentration (*Huang et al., 2022*). However, contrary to our results, external sodium selenite can significantly induce the expressions of genes involved in flavone and flavonol biosynthesis (*Li et al., 2020*). The current study's results deviate from previous research and emphasize the significance of employing proper selenite concentrations for treatment. Appropriate sodium selenite applications can increase the levels of selenium and other active chemicals in eukaryotic organisms; however, it may have the opposite effect after surpassing a specific concentration threshold.

## CONCLUSIONS

In the food industry, selenium-enriched mushrooms play a significant role as a daily supplement. Therefore, RNA-Seq was used to evaluated the effect of different concentrations (5 μg mL$^{-1}$ and 10 μg mL$^{-1}$) of Se on the gene expression changes of *M. esculenta.* Both treatments can cause the expression level changes of genes involved in cell cycle, meiosis, aminoacyl-tRNA biosynthesis, spliceosome, mRNA surveillance pathway, and protein processing in endoplasmic reticulum pathway. Among which, one MFS substrate transporter and one aspartate aminotransferase genes were significantly upregulated and were presumed to be involved in Se metabolism. Moreover, the level of gene expression changed increased with selenite treatment concentration, suggesting that the high levels of Se promoted the detoxification of *M. esculenta*. However, genes related to isoflavone biosynthesis and flavonoid metabolism were significantly downregulated under selenite stress, highlighting the importance for selenite treatment with appropriate concentrations to obtain the balance of Se content and other active compounds. The results may serve as a basis for elucidating the specific effects of Se on growth and accumulation/detoxication processes in *M. esculenta* and promoted functional selenium-enriched mushroom production.

### Funding
This research was funded by the Natural Science Foundation of Anhui Provincial Department of Education (KJ2021ZD0107), the University Synergy Innovation Program of Anhui Province (Grant No. GXXT-2023-054), the Special Project of Functional Agriculture of Anhui Science and Technology University (2021gnny06), the Key projects of Anhui Provincial Education Department (KJ2021A0881) and the University students' innovation and entrepreneurship projects (202210879097). The funders had no role in study design, data collection and analysis, decision to publish, or preparation of the manuscript.

### Grant Disclosures
The following grant information was disclosed by the authors:

Natural Science Foundation of Anhui Provincial Department of Education: KJ2021ZD0107.
The University Synergy Innovation Program of Anhui Province: GXXT-2023-054.
Special Project of Functional Agriculture of Anhui Science and Technology University: 2021gnny06.
The Key projects of Anhui Provincial Education Department: KJ2021A0881.
University students' innovation and entrepreneurship projects: 202210879097.

## Competing Interests

The authors declare there are no competing interests.

## Author Contributions

- Mengxiang Du performed the experiments, prepared figures and/or tables, and approved the final draft.
- Shengwei Huang conceived and designed the experiments, authored or reviewed drafts of the article, and approved the final draft.
- Zihan Huang performed the experiments, prepared figures and/or tables, and approved the final draft.
- Lijuan Qian performed the experiments, analyzed the data, prepared figures and/or tables, and approved the final draft.
- Yang Gui analyzed the data, prepared figures and/or tables, and approved the final draft.
- Jing Hu analyzed the data, prepared figures and/or tables, and approved the final draft.
- Yujun Sun conceived and designed the experiments, authored or reviewed drafts of the article, and approved the final draft.

## DNA Deposition

The following information was supplied regarding the deposition of DNA sequences:

The raw sequences for each sample are available at NCBI Sequence Read Archive (SRA): PRJNA1006517.

## Data Availability

The raw sequences for each sample are available at NCBI Sequence Read Archive (SRA): PRJNA1006517.

## Supplemental Information

Supplemental information for this article can be found online at http://dx.doi.org/10.7717/peerj.17426#supplemental-information.

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
