# Peer review of "De novo assembly and characterization of the transcriptome of Morchella esculenta growth with selenium supplementation"

_PeerJ, doi:10.7717/peerj.17426_

## Round 0.1 · original submission · Minor Revisions

Please revise the manuscript by following the reviewers' suggestions and comments.

Reviewer 1 ·

Basic reporting

The manuscript is readable well and most parts are clear, unambiguous and use of terminus technicus correctly, but some sections are not enough detalied and uncleared. It is required to check the grammar and the types of fonts (e.g., Latin names are italics, L85, L100, etc.) in the entire text and the references because it contains many mistakes.

The article has 44 different references that describe the background of this study. The literature used is quantitatively and qualitatively adequate, and the line of thought introduced in the introduction fits the manuscript and topic. My opinion is the chapter of introduction is longer than as usual, it is approximately 4.5 pages, it must be written shorter. I suggest the part of the selenium effects on human health can be shorter (My suggestion is some statements transferring to the discussion section, e.g., L60-64). Because the article focuses on the M. esculenta transcriptomic analysis during different amounts of Se accumulations. The parts of the Se-accumulation in mushrooms in the text are valuable and detailed. However, the introduction does not show any potential selenium sources and examples (Se enriched foods, plants, etc.).

The abstract is well-written and appropriate to standards (length 199 words), but it contains a few mistakes, The M. esculenta is not a commonly grown mushroom, because it only fruits under hardwoods and conifers in a short period (depending on the habitat 3-5 months). The usage of capital letters is not appropriate in the text (e.g., L20-21). Abbreviations of MSe5 and MSe10 should be explained in the text the first time.

The numbers and labels are readable and clear in Figure 1. The Volcano plot in Figure 2 is not too informative because there are lots of points, but the colour indication is very useful. The Venn-diagram shows the same result, which is more understandable. I suggest you make two or three Venn diagrams that show the up-regulated, down-regulated, and non-differentially expressed genes. I suggest changing Figure 3 to a table because it contains many repetitions among the gene functions, and the comparison of the data about up- and down-regulated genes can be even more easily, respectively. The letters in the Table 1. are not mentioned in the caption.

The data is available in NCBI, I found RNA seq data of Morchella esculenta, these are packed in 9 different files in sp. file format. The submitted manuscript represents an appropriate amount of relevant information for the publication and all results support the hypothesis.

Experimental design

The topic of the manuscript is relevant to PeerJ, it contains mycology and bioinformatics. In terms of its subject, the manuscript is appropriate for the publication in PeerJ.

The research questions focus on the aspects of the selenite uptake and tolerance and the role of the different metabolic pathways in the M. esculenta. Moreover, it addresses dose-specific lesser-known transcriptomic mechanisms, too. This research will be a great base for future research on the role of selenium metabolism in the Ascomycetous macrofungi. During the investigation, used methods and materials have been conducted to high technical standards.

I found a mistake, because the text contains the Illumina Hiseq 4000 platform at Shanghai Meiji Biological Medicine Technology Co., Ltd while the description of the NCBI database contains Illumina NovaSeq 6000 sequencer at Shanghai BIOZERON Co., Ltd.

The methodolgy has been raised questions and some parts are not enough detailed.

“The origin of the ACCC50764 is from Agricultural Culture Collection of China (Beijing, China)”.

Was the PDA medium made commercial powder or prepared in-house?

A short list of analytical grade reagents and molecular kits is needed.

What were the units of the growth rates obtained (mm, cm, %)?

L158 - hydride generation-atomic fluorescence spectrometry

What mass of mycelium was collected from each sample? Based on the biomass results, the growth of mycelia slowed down significantly at the high-dose selenium concentration.

Were the quality and quantity of RNA extraction measured with Nanodrop?

How were differentially expressed genes (DEGs) randomly selected?

The sections of Genes involved in Se enrichment or biotransformation and Conclusion have different spacing than the early body text.

Validity of the findings

The authors mapped the transcriptome of M. esculenta, furthermore the identified the up- (800) and down-regulated (13,355) genes during low and high selenite treatments. 475 (high) and 2,381 (low) genes were selected which are expressed comparing the control.

The section of Conclusions summarize the manuscript well and present appropriate conclusions, however, the limitations of the experiments are not detailed enough.

Reviewer 2 ·

Basic reporting

Dear author/authors;
The study titled 'De novo assembly and characterization of the transcriptome of Morchella esculenta growth with selenium supplementation‘ is comprehensive and will contribute to the literature. However, there are some shortcomings in the writing of the article, which are stated below. An attempt has been made to indicate the deficiencies using line numbers. The entire article should be checked. It should be taken into consideration that some of the errors that are noticeable throughout the article may also occur in lines not included below. For example; “Mse5/MSe5”, “Mse10/MSe10”, “down-regulated” or “downregulated”. Abbreviations and technical terms should be used in the same way throughout the article, and abbreviations that are not written in long form should be written long in the first place they are used. Finally, it was observed that some species names were not written appropriately. This should be noted throughout the article.
Best regards.

Experimental design

The experimental design of the study meets the journal standards. With this, defining information and methods regarding the method of determining colony color are not included in the "materials and methods".

Validity of the findings

In this section, the results are presented in a very effective and readable manner. However, discussion with the literature is insufficient, especially in the early parts of the results and discussion. As can be seen throughout the manuscript, there are studies conducted using selenium. It is necessary to examine the studies in the literature, rewrite the discussion sections and make references to new/related studies (especially research articles).

Additional comments

The deficiencies and those that need to be corrected are included in the attached file.

Annotated reviews are not available for download in order to protect the identity of reviewers who chose to remain anonymous.

---

## Round 0.2 · Minor Revisions

The manuscript is provisionally acceptable based on the reviewers' comments and decisions. However, there are still some issues that need to be addressed before the manuscript can be fully accepted for publication.

Reviewer 1 ·

Basic reporting

The article was written readable, clear and using of terminus technicus correctly, the corrections can be tracked. Hereby thank you that you took my and the other reviewer suggestions and advices.

The abstract is well-written and appropriate to standards (length 198 words), in addition, the mistakes were corrected. The keywords also were corrected.
The shorter introduction version is more readable, consistent and more focused for the topics of molecular biology and Se accumulation in the mushrooms. The marking of the Latin names is adequate.

The section of the Materials and methods is improved and good enough detailed. The section of the Results is more detailed and the interpretation is clearer and well-derived to conclusions.

I found a few misspelled words, but it is not serious, I have a list for you:
• L18 – in both selenium-treated groups
• L26 – Keywords: M.
• L75 - Transcriptional profiling
• L80 – double end-of-sentence points
• L84 – Morchella esculenta (commonly)
• L89 – anti-inflammatory (Akyuz…)
• L110 – 26±0.5
• L124 – Erlenmeyer
• L125 – Morchella
• L155 – short-read assembly
• L204 – double end-of-sentence points
• L258, 261, 264 – binding (214 DEGs) (see
• L307 – genes (doubled)
• L314 – CK and MSe10
• L377 – ). Therefore

Experimental design

The research well defined.

Validity of the findings

The conclusions are well stated.

Reviewer 2 ·

Basic reporting

Dear editor,

The authors have made the anticipated corrections. A few deficiencies/errors were identified in the revised manuscript. There is no problem in publishing it after these corrections are made.

Best Regards.

Introduction
Line 86: Need to cite relevant research articles (end of the sentence).
Line 90: The reference must be written in accordance with the rules. Also, need some new reference about anti-bacterial and anti-inflammatory effects.

Experimental design

Materials & Methods
Line 135: Need to a space/gap after number 26.

Validity of the findings

Results and Discussion
Line 307: I think the word "genes" was written twice.
Line 314: There is a more space/gap after word “and”.

Annotated reviews are not available for download in order to protect the identity of reviewers who chose to remain anonymous.

---

## Round 0.3 · accepted · Accept

The authors have addressed the reviewer's questions point-by-point.